# Optimization of High Temperature-Resistant Modified Starch Polyamine Anti-Collapse Water-Based Drilling Fluid System for Deep Shale Reservoir

**DOI:** 10.3390/molecules27248936

**Published:** 2022-12-15

**Authors:** Xiangwei Kong, Mingzhong Chen, Chaoju Zhang, Zuocai Liu, Yanxin Jin, Xue Wang, Minggang Liu, Song Li

**Affiliations:** 1School of Petroleum Engineering, Yangtze University, Wuhan 430100, China; 2Hubei Key Laboratory of Oil and Gas Drilling and Production Engineering, Yangtze University, Wuhan 430100, China; 3Downhole Operation Company, CNPC Chuanqing Drilling Engineering Company, Chengdu 610000, China; 4Sinopec Southwest Petroleum Engineering Co., Ltd., Chengdu 610100, China; 5Chongqing Drilling Company, Sinopec Southwest Petroleum Engineering Co., Ltd., Chongqing 400042, China; 6Sinopec Qingdao Safety Engineering Research Institute, Qingdao 266000, China; 7Engineering Research Institute, PetroChina Southwest Oil and Gas Field Company, Chengdu 610017, China

**Keywords:** deep shale reservoir, horizontal well, anti-collapse, water-based drilling fluid, modified starch, filtration loss

## Abstract

During drilling in deep shale gas reservoirs, drilling fluid losses, hole wall collapses, and additional problems occur frequently due to the development of natural fractures in the shale formation, resulting in a high number of engineering accidents such as drilling fluid leaks, sticking, mud packings, and buried drilling tools. Moreover, the horizontal section of horizontal well is long (about 1500 m), and the problems of friction, rock carrying, and reservoir pollution are extremely prominent. The performance of drilling fluids directly affects drilling efficiency, the rate of engineering accidents, and the reservoir protection effect. In order to overcome the problems of high filtration in deep shale formations, collapse of borehole walls, sticking of pipes, mud inclusions, etc., optimization studies of water-based drilling fluid systems have been conducted with the primary purpose of controlling the rheology and water loss of drilling fluid. The experimental evaluation of the adsorption characteristics of “KCl + polyamine” anti-collapse inhibitor on the surface of clay particles and its influence on the morphology of bentonite was carried out, and the mechanism of inhibiting clay mineral hydration expansion was discussed. The idea of controlling the rheology and water loss of drilling fluid with high temperature resistant modified starch and strengthening the inhibition performance of drilling fluid with “KCl + polyamine” was put forward, and a high temperature-resistant modified starch polyamine anti-sloughing drilling fluid system with stable performance and strong plugging and strong inhibition was optimized. The temperature resistance of the optimized water-based drilling fluid system can reach 180 °C. Applied to on-site drilling of deep shale gas horizontal wells, it effectively reduces the rate of complex accidents such as sticking, mud bagging, and reaming that occur when resistance is encountered during shale formation drilling. The time for a single well to trip when encountering resistance decreases from 2–3 d in the early stages to 3–10 h. The re-use rate of the second spudded slurry is 100 percent, significantly reducing the rate of complex drilling accidents and saving drilling costs. It firmly supports the optimal and rapid construction of deep shale gas horizontal wells.

## 1. Introduction

Shale gas is a pure and high-quality natural gas resource that is widely distributed in the world and has abundant reserves. The technical recoverable resources of shale gas in China are about 21.8 × 10^12^ m^3^, with huge resource value and social value. Sichuan Basin is currently the most favorable shale gas exploration and development area in China [1,2,3,4,5,6,7,8,9]. During drilling in deep shale formation, due to the development of natural fractures and joints in shale formation, problems such as lost circulation and collapse occur easily during drilling in long horizontal section (about 1500 m), resulting in a large number of engineering accidents such as drilling fluid loss, drill pipe sticking, and drilling tool burying [10,11,12]. The horizontal section of the shaft is very long, and the problems of friction, rock-carrying, and formation contamination are extremely prominent. The quality of the drilling fluid directly affects the drilling efficiency, the incidence of engineering accidents, and the effect of reservoir protection [13,14,15]. Due to the development of natural fractures and joints in shale sections, the performance of drilling fluids is compromised due to severe filtration, resulting in complex downhole accidents such as mud bags, drill pipe sticking, and wall collapse. The effects of high temperatures on water-based drilling fluids are complex and varied. High temperature is commonly considered as an essential factor leading to various components of drilling fluid and their physical and chemical interactions [16]. Among them, high temperature is the basis for clay action in drilling fluid, and the role of treating agent is the key [17]. The effects on clay mainly include high temperature dispersion, high temperature coalescence, and high temperature activation of clay particles. Among them, high temperature dispersion of clay refers to the automatic dispersion of clay particles, especially bentonite particles, in a drilling fluid under the action of high temperature. Montmorillonite has the strongest hydration and dispersal capabilities. At high temperatures, the diversity of montmorillonite particles increases, the concentration of particles increases, and the specific surface area increases. Macroscopically, the apparent viscosity (AV), static shear force (GS), and dynamic shear force (YP) also increase [18]. High temperature coalescence not only decreases the quality of the drill fluid filtration cake, but also increases the HTHP filtration of the drill fluid. High temperature activation of clay particles refers to the phenomenon that the surface activity of clay particles decreases after the clay suspension is subjected to high temperature (normally higher than 130 °C). High temperature passivation of clay is a reversible permanent shift that does not shift with temperature, reducing the unit adsorption capacity on the surface of clay particles, and thus affecting the thermal stability of the drilling fluid. The effects of high temperatures on polymer drilling fluid processing agents are mainly reflected in two aspects: high temperature degradation and high temperature cross-linking. High temperature degradation is the breaking of molecular chains of macromolecular organic compounds by high temperatures. There are two scenarios for the drilling fluid treatment agent: the fracture of the polymer backbone and the breaking of the bond between the hydrophilic group and the backbone. The factors affecting high temperature degradation include molecular structure and external conditions (such as temperature, system oxygen content, system salt content, pH value, shear action, action time, etc.), of which molecular structure is the primary factor [19]. pH values and action times are essential factors that affect the high temperature degradation of drilling fluid treatment agents in the external environment. The higher the pH of the drilling fluid, the faster the degradation will occur, and the rate of degradation of the high temperature molecules will become faster as the high temperature action time becomes longer. The performance of the drilling fluid changes with increasing temperature, which is mainly reflected in three aspects: the deterioration of the drilling fluid performance, the rheological properties of the drilling fluid, and the wall-building properties of the drilling fluid filtration [20]. The influence of high temperature on the rheological property of drilling fluid is relatively complex, and its specific conditions can be divided into the following three types according to the relationship between viscosity and temperature: the viscosity of drilling fluid decreases with the increase of temperature, the viscosity of drilling fluid increases with the increase of temperature, and the viscosity of drilling fluid decreases first and then increases with the increase of temperature [21]. It is common for the filtration loss of drilling fluid to increase and the thickness of the filter cake to increase after high temperatures. The deeper the well, the higher the temperature and the more severe the filtration.

In order to meet the challenges brought by the high temperature and high pressure environment of deep well and ultra-deep well formations, it is necessary to modify the original fluid loss agent products (such as modified starch, modified humic acid, modified phenolic resin, etc.) or artificially synthesize the current high temperature fluid loss agent.

Starch-modified fluid loss agents are economical, abundant in sources, non-toxic, and easy to degrade, but they are less resistant to temperature. Only a few products have temperature resistance above 130 °C [22,23,24]. As one of the first fluid loss reducers for drilling fluids, starch is structurally similar to cellulose and belongs to the carbohydrate group. Moreover, the additional chemical properties of starch are similar to those of cellulose, which can also undergo etherification, esterification, grafting, carboxymethylation, and cross-linking reactions to obtain a range of modified products. In the 1990s, Audibert et al. [25] developed a fluid loss additive suitable for a high temperature and high pressure water-based drilling fluid system. The product is a sulfonated styrene derivative of the current type with strong high temperature degradation resistance. Because of this high-temperature degradation resistance, it effectively prevents water-based drilling fluids from damaging the oil layer. After aging the aqueous solution system at 160 °C, there is no gelling. It is shown that the agent is thermally stable. Drilling fluid systems prepared with fluid loss reducers have also shown superior fluid loss reduction performance in dynamic filtration experiments. Patel AD et al. [26] used AMPS as the polymerization monomer and *N,N*′-methylene bisacrylamide (MBA) as the cross-linking agent to synthesize a water-based drilling fluid loss reducer with high temperature resistance through controlled cross-linking. The agent has superior resistance to temperature and excellent resistance to calcium. Soric and Heier [27,28] synthesized a new generation of high temperature fluid loss additive with temperature resistance up to 230 °C, using vinyl amine (VA) and vinyl sulfonic acid (VS) as the main reaction monomers. Rupinski et al. (2009) studied the influence of modified starch and their blends with tylose as protective agents in the filtration of drilling fluids, as well as replacement of tylose by modified starch, were investigated [29]. Dias et al. (2015) investigated the potential of using starch derivatives modified with vinyl esters from fatty acids, as additives to control filtrate in invert-emulsion (W/O) drilling fluids [30]. They evaluated the synthetic drilling fluids by means of physical–chemical filtration tests at high temperature and pressure, along with electrical stability and rheological tests, which indicated that the formulations developed from fatty esters from starch were able to compete technically with the standard drilling fluid. Soto et al. (2020) evaluated the starch (NCS) as a filter control agent in fresh water-based drilling fluids, which were subjected to high pressure and high temperature filtering [31]. Zhong et al. (2021) investigated the potential of utilizing starch nanospheres (SNSs) for water-based drilling fluid, and the effect of SNSs on the rheological and filtration properties of bentonite-based drilling fluid was investigated [32]. Sulaimon et al. (2021) tried the modified starches from cassava and maize to use for enhancing the properties of water-based muds under high pressure high temperature (HPHT) conditions and comparing their performances with that of the CMC. They proved experimentally that these modified starch-added muds could replace CMC as fluid loss agents, which can withstand HPHT conditions [33]. Long et al. (2022) experimentally tested the filtration loss performance of CMCS before and after hot-rolling aging at 120 °C in 4.00% NaCl and saturated NaCl brine base slurry. The experimental results confirm that CMCS have a better tolerance to high temperature of 120 °C and high concentration of NaCl [34].

The above researchers have done a lot of experimental research work on CMC and plant starch as filtrate reducer of water-based drilling fluid system and have also generated a lot of research results. However, the experimental temperature is mostly limited to 140 °C, and only modified starch cannot fully meet the requirements of drilling fluid filtration reduction. Other additives should also be considered to be used together to achieve strong inhibition and plugging of drilling fluid filtration, so as to further enhance the performance and enhance the properties of water-based muds applied for deep high pressure high temperature (HPHT) shale gas reservoirs.

To overcome the problems of high filtration in deep shale formations, collapse of borehole walls, sticking of pipes, mud inclusions, etc., optimization studies of water-based drilling fluid systems have been conducted with the primary purpose of controlling the rheology and water loss of the drilling fluid. The experimental evaluation of the adsorption characteristics of “KCl + polyamine” anti-collapse inhibitor on the surface of clay particles and its influence on the morphology of bentonite was carried out, and the mechanism of inhibiting clay mineral hydration expansion was discussed. The idea of controlling the rheology and water loss of drilling fluid with high temperature resistant modified starch and strengthening the inhibition performance of drilling fluid with “KCl + polyamine” was put forward, and a high temperature resistant modified starch polyamine anti-sloughing drilling fluid system with stable performance and strong plugging and strong inhibition was optimized. This optimized water-based drilling fluid system uses modified starch as the main filtration reducing agent and “KCl + polyamine” as the main inhibiting agent. It features high-temperature resistance, environmental protection, and strong anti-pollution capabilities. It meets the performance requirements for drilling fluids with low filtration, strong plug, strong inhibition, and stable properties, and effectively addresses complex downhole problems such as drilling clogs and mud bags in shale formations.

## 2. Development of a Polyamine Anti-Sloughing Inhibitor

### 2.1. Polyamine Inhibits the Adsorption of Anti-Sloughing Agents on the Surface of Clay Particles

Thermal analysis is a method to determine the state change of a substance based on changes in its properties due to temperature shifts, such as thermal energy, mass, size, structure, etc. It is widely used for scientific research and industrial production in various fields. In numerous experiments, the adsorption of organic matter onto matter has been measured indirectly by UV-visible spectrophotometry (Figure 1). However, UV-visible spectroscopy can only measure organic compounds with conjugated double bonds. Due to the absence of conjugated double bonds in polyamine, the adsorption of polyamine onto clay cannot be determined by UV-visible spectroscopy. In the present experiment, the TG-DTA technique was used to determine the amount of polyamine adsorbed by the clay.

It can be seen from Table 1 that the adsorption rate of the polyamine shale inhibitor on the clays evaluated in the laboratory is between 66 and 74. It can be shown that polyamine inhibiting anti-sloughing agents can be adsorbed on the clay surface to prevent hydration, expansion and dispersion of the clay.

### 2.2. Analysis of the Adsorption Properties of Polyamine Inhibitors and Anti-Sloughing Agents

Since the composition of bentonite is Al_2_O_3_ 4SiO_2_ H_2_O + nH_2_O, the front H_2_O is structural water (crystal water), and the back nH_2_O is adsorbed water, their infrared spectra are different. The theoretical molecular formula of montmorillonite is (OH) 4Si_8_Al_4_O_20_. Its unit cell system consists of two layers of silicon-oxygen tetrahedra sandwiched by a layer of aluminum–oxygen octahedra arranged in parallel. Al^3+^ in octahedron of montmorillonite and (or) Si^4+^ in tetrahedron are frequently partially (or entirely) replaced by additional cations to form some chemical varieties of bentonite. The absorption peaks of Si-O bond stretching vibrations and Al-O vibrations of silicon oxide tetrahedra are the most prominent features of bentonite at wave numbers 1000–1100 cm^−1^ in the infrared spectrum. Without this absorption peak, it has no crystalline structure. The basic frequency oscillations and the main absorption peaks corresponding to the IR spectral activity of the sodium bentonite structures are listed in the Table 2 below.

Infrared spectroscopy was performed on the polyamine-modified Na_3_ as shown in Figure 2. It shows that the skeleton of bentonite is unchanged after polyamine modification. At the same time, different absorption peaks appear in the IR spectra of polyamine-modified soil, where 2926.16 cm^−1^, 2918.40 cm^−1^, 2914.67 cm^−1^, and 2976.27 cm^−1^ are elongated vibrational peaks of methylene C-H, indicating that polyamine has been adsorbed onto the clay surface. In addition, the bending vibration absorption peak of water molecules at 1637.88 cm^−1^ in sodium bentonite is weakened (Figure 3), indicating that polyamine enters into the sodium bentonite soil layer and extrudes some water molecules from the interlayer, resulting in the reduction of interlayer water content.

### 2.3. Effect of Polyamine Anti Sloughing Agent on the Surface Morphology of Bentonite

SEM (Hitachi S-4800) was used to detect the influence of water and polyamine inhibitor on the surface morphology of bentonite and analyze their inhibition mechanism on clay. A measure of 30 mL of 1.0% polyamine solution and 30 mL of pure water, each containing 8 g of bentonite, were left to stand overnight and dried in 105 ovens for 10 h, and the secondary bentonite was selected as clay. The results are shown in Figure 4 and Figure 5.

Montmorillonite consists of a T-O-T layered structure consisting of two layers of Si-O tetrahedra sandwiched by one layer of A1-O octahedra. When there is no water or only a small amount of water molecules in the crystal layers of montmorillonite, the crystal layers are connected by a weak van der Waals force. When the montmorillonite is observed by electron microscope (SEM), the photos shown in Figure 5 can be obtained. The sheet distribution of montmorillonite is clearly shown in Figure 5, with stratified pores between the sheets. In addition to this, it will be seen that, with the exception of a very small number of strata, the montmorillonite is of uniform thickness, and arranged in an orderly manner. After immersion in polyamine, the interlayer spacing of montmorillonite is considerably increased by the insertion of polyamine, whose polarity destroys the original van der Waals force between the layers.

### 2.4. Mechanism of Action of Polyamine Anti-Collapse Inhibitors

There are three ways to suppress hydration expansion in clay, namely, ion exchange, coating of clay particles, and reduction of the hydrophilicity of clay surfaces. Due to the small hydration radius and low hydration energy, potassium ions easily enter the space between the two oxygen hexagons of the clay layer to exchange cations between layers, thus inhibiting hydration expansion of the clay. Polyamine, an anti-collapse inhibitor, inhibits hydration expansion in clay. The suppression mechanism is that the amine nitrogen atoms of the polyamine shale inhibitor do not share electron pairs and can bind to protons. As a result, when the polyamine shale inhibitor is dissolved in water, it picks up protons from the water and forms positively charged ammonium ions, making the aqueous solution weakly alkaline. Its hydrolysis formula is as follows:(1)$${\mathrm{R-NH}}_{2}\;+\;H_{2}O\;⇌\;{{\mathrm{R-NH}}_{3}}^{+}\;+\;{\mathrm{OH}}^{-}$$

Protonated ammonium ions neutralize the negative charge on the surface of clay particles by electrostatic action, reducing the hydration repulsion between clay layers in a manner similar to that of potassium ions. The isoelectric points of clay particles are normally 6–8. When the pH value of the system is larger than the isoelectric point, the end face and the surface of the clay particle are negatively charged. Low molecular weight polyamine molecules partially dissociate in solution to form ammonium orthorhombic ions, which form a chemical potential difference with inorganic cations between the clay layers. Driven by the chemical potential difference, polyamines enter the clay layer and the protonated ammonium ions displace the inorganic hydration cations through ion exchange, reducing the Zeta potential of the clay particles. In addition, the amino groups in its molecule bind to protons to generate two ammonium ortho ions, which are adsorbed separately on adjacent clay layers, binding the clay layers together. Therefore, the adsorption of polyamines on clay surfaces is an irreversible process and is not easily desorbed by additional ion exchange. At the same time, electrostatic attraction and hydrogen bonding work together to reduce the interlayer hydration repulsion of the clay, binding the clay layers together and squeezing some of the interlayer adsorbed water side by side to weaken the hydration of the clay. After the polyamine inhibitor anti-sloughing agent forms a monolayer adsorption on the surface of clay particles, the hydrophobic groups on the molecular chain of polyamine inhibitor anti-sloughing agent partially cover the clay surface, changing the structure of the clay surface, reducing the hydrophilicity of the clay, enhancing the hydrophobicity, preventing the entry of water molecules, and further inhibiting the hydration expansion of the clay.

## 3. High Temperature Resistant Modified Starch

Polymerized monomers of vinyl were developed, and starch was modified by grafting copolymerization. High temperature-resistant modified starch was prepared and a modified starch drilling fluid system was formed from the high temperature-resistant modified starch. The indoor experimental evaluation shows that the modified starch with high temperature resistance can resist the temperature of 180 °C, has excellent water loss reduction effect, and has the characteristics of high temperature resistance, environmental protection, and anti-pollution.

### 3.1. Action Mechanism of Modified Starch

A novel high-performance polymer monomer was designed and synthesized. By grafting starch with this monomer, a modified starch material with excellent properties was obtained. This has excellent high temperature resistance; the temperature resistance can reach more than 180 °C. It also has excellent salt and calcium resistance and inhibition performance. There are three main reasons for the apparent effectiveness of modified starch.

① In the design and synthesis of monomer, the balance between the molecular weight of monomer and the group is completely considered. On the basis of satisfying the necessary groups, the molecular weight of the monomer should not be too large, so that the monomer can maintain excellent polymerization activity. As a result, monomeric grafting modified starch has higher grafting yield and grafting efficiency.

② Special structures in the monomer will produce ring structures after polymerization, and these ring structures have a great contribution to the high temperature resistance. By increasing steric hindrance, the desorption of polymers onto the clay at high temperatures is suppressed.

③ The graft copolymer structure contains additional hydration and adsorption groups, which will form strong adsorption with clay, shift the size distribution of clay particles, so as to form thin and dense filter cake, and achieve the purpose of reducing the filtration loss at high temperature and pressure. In general, the excellent properties of polymeric monomers give graft copolymers good synthesis properties.

### 3.2. Evaluation of Temperature Resistance of Modified Starch

According to the national standard of China: GB/T 16783.1-2014, Petroleum and natural gas industries—Field testing of drilling fluids—Part 1: Water based drilling fluids, the evaluation method for the temperature resistance of the fluid loss reducer is: measuring a certain amount of water-based drilling fluid base fluid, adding 3.0% modified starch to reduce the fluid loss, and the drilling fluid is subject to rolling aging at 120 °C, 160 °C, 180 °C, and 200 °C for 16 h, and measuring the fluid loss.

It can be seen from Figure 6 that when the experimental temperature is 120–180 °C, the filter loss does not exceed 12 mL, and the increase of the filter loss is relatively slow with the increase of the temperature. However, when the temperature reaches 200 °C, the filter loss exceeds 12 mL, and the increase trend of the filter loss is stronger than that of the previous filter loss. Therefore, the analysis shows that the filter loss at this temperature is not up to the standard, indicating that the performance of the additive is poor when the temperature is 200 °C, so the temperature resistance of this material can reach 180 °C.

### 3.3. Evaluation of Salt Resistance of Modified Starch

Evaluation of salt resistance: 350 mL of freshwater base slurry was measured, a certain amount of NaCl was added, stirred for 5 min at high speed, then 2.0% modified starch was added, stirred for 5 min at high speed, and the medium pressure filtration loss was measured after curing at room temperature for 24 h. The shift in the filtration loss of the modified starch was identified by investigating the increase of NaCl in the freshwater base pulp. The lower the filtration loss, the better.

As can be seen in Figure 7, the amount of NaCl gradually increases when a fixed amount of modified starch of 2.0% is used, and the filtration loss is not gradually increasing but unusually gradually decreasing. The ability of modified starch to exhibit such salt resistance is primarily due to its structure. Modified starch is a zwitterionic polymer with virtually equal positive and negative charges. In pure water or aqueous solution with low salt content, due to the electrostatic attraction between molecules, the molecular chains are curled, and the sulfonic acid groups contained cannot be completely exposed to interact with the clay, forming a thin hydration film and the surface of clay particles ζ. The absolute value of the potential is not very high, resulting in its hydrodynamic loss-reduction performance not being fully exploited. In aqueous solutions with high salt content, the presence of tiny molecular inorganic salts shields the electrostatic attraction between the polymer molecules, which is then converted into an electrostatic repulsion between the molecules. In contrast, the polymer molecular chain is more extended, the sulfonic acid group is fully exposed, the viscosity of the system is increased, and the filtration reduction performance is improved. As the salt content is gradually increased, the filtration reduction properties of the modified starch are gradually enhanced.

### 3.4. Evaluation of Salt Resistance of Modified Starch

Evaluation method: 350 mL saturated saline base slurry was measured, a certain amount of CaCl_2_ was added and stirred at high speed for 5 min, then 1.5% modified starch was added and stirred at high speed for 5 min, and the medium pressure filtration loss was measured after curing at room temperature for 24 h. The amount of CaCl_2_ in the saturated brine slurry was continuously increased and the shift of the filtration loss was measured (Figure 8).

## 4. High Temperature-Resistant Modified Starch KCl Polyamine Anti-Collapse Drilling Fluid System

### 4.1. Drilling Fluid Formula

According to the experiment on the compatibility of high temperature-resistant modified starch, polyamine inhibitor, and anti-collapse agent with potassium chloride, the experiment on the compatibility of polyamine with polymer treatment agent, and the experiment on the compatibility of polyamine with sulfonated treatment agent, the polymer treatment agent and sulfonated treatment agent that are well compounded with high temperature resistant modified starch and polyamine inhibitor and anti-collapse agent are selected, and the optimal dosage of the polymer treatment agent and sulfonated treatment agent with modified starch and polyamine inhibitor and anti-collapse agent is obtained. The formula of high temperature resistant modified starch polyamine anti-sloughing drilling fluid system is determined as follows:

2–3% Bentonite + 0.3–0.5% polyanion cellulose + 0.3–0.5% amphoteric polymer coating agent + 0.3–0.5% high temperature resistant modified starch + 0.3–0.5% polyamine + 0.5–0.8% sulfonated phenolic resin + 2–4% modified asphalt + 1–2% film forming plugging agent + 5–7% potassium chloride + 2–3% temperature-resistant saturated salt lubricant + 1–2% ultra-fine calcium carbonate.

### 4.2. Routine Performance Evaluation of Drilling Fluid System

The rheological properties, water loss properties, and inhibition properties of the above formula (increased to 1.75 g/cm^3^) were measured at different aging temperatures (80–160 °C) and aging times. The specific results are shown in Table 3. According to the data analysis in Table 3, the rheological property of the system is relatively stable after aging at 80–160 °C for different aging times. After aging at 160 °C for 64 h, the rheological property and high temperature and high pressure water loss began to increase significantly, and the plastic viscosity and dynamic shear force decreased significantly, indicating that the system has excellent high temperature stability.

It can be seen from the comparison that after aging at 80–160 °C for different aging times (Table 3), the addition of polyamine inhibitor polyamine in the base slurry significantly improves the inhibition performance of the system. The inhibition performance of the polyamine potassium chloride system is also excellent compared to that of the potassium chloride base slurry system, with rock cuttings recovering more than 97 percent, indicating the favorable suppression performance of the system. It can be deduced from the analysis of the linear expansion rate data of the system that the polyamine potassium chloride system still has strong inhibitory properties and can meet the requirements of field operations.

### 4.3. Evaluation of Salt Resistance and Calcium Resistance

Adding different amounts of sodium chloride into the system (weighing up to 1.75 g/cm^3^), we measured its rheology, water loss, and inhibition performance, and analyzed the salt resistance of the system. As can be seen from Table 4, the plastic viscosity and dynamic shear forces of the polyamine potassium chloride system increased with the addition of sodium chloride, while the water loss decreased and the suppression performance was relatively stable. When the addition of sodium chloride reached 10% (the concentration of Cl^−^ was 100,000), the rheological property and water loss shifted slightly, indicating that the system has excellent salt resistance.

We added different amounts of calcium chloride into the system (weighing up to 1.75 g/cm^3^), measured its rheology, water loss, and inhibition performance, and analyzed the calcium resistance of the polyamine potassium chloride system. It can be concluded from Table 5 that with the increase of calcium chloride dosage in the polyamine potassium chloride system, the plastic viscosity and dynamic shear force of the system showed an upward trend, the filtration loss changed little and remained at a low level, while the inhibition decreased slightly. From the overall situation, the system has strong calcium pollution resistance, and still has excellent performance under the dosage of 1.6% calcium chloride (the concentration of Ca^2+^ is 5500 ppm), The calcium resistance of the system meets the contract requirements.

### 4.4. Evaluation of Anti-Poor Soil Pollution

The reservoir rock was ground to form 100 mesh particles and then added to the polyamine potassium chloride drilling fluid system (increased to 1.75 g/cm^3^). After aging at 135 °C for 16 h, the system’s resistance to soil pollution was investigated and the system performance changes were observed. The results of the experiments are given in the following Table 6.

According to the experimental data in Table 6, the plastic viscosity, dynamic shear force, and initial and final shear of the system showed a significant upward trend, while the filtration loss showed a downward trend, with the addition of soil from 5% to 20%. This is because the viscosity and shear forces of the system increased after the fine-grained soil was dispersed into the polyamine potassium chloride system, while the filling of the fine-grained soil into the filter cake pores reduced the mud water loss. Experimental results show that the system has excellent resistance to soil contamination.

Based on comprehensive analysis, the polyamine potassium chloride drilling fluid system was optimized through system formulation and anti-pollution performance evaluation. The system has strong inhibition performance, stable performance, excellent indoor temperature resistance up to 150 °C, anti Cl^−^ pollution ability up to 100,000 ppm, anti Ca^2+^ pollution ability up to 5500 ppm, and anti-poor soil pollution ability up to 15% at 1.75 g/cm^3^.

### 4.5. Application

Small scale drilling fluid performance test was carried out on site. After the optimized drilling fluid was fully circulated for three weeks, the test parameters of drilling fluid performance test are shown in Table 7, and all indicators meet API standards. The effect of reducing HTHP filtration of drilling fluid is obvious, the quality of mud cake is improved, and the mud cake is smooth, dense, and flexible, which has little impact on the rheology of drilling fluid.

It can be concluded from the Table 7 that after adding modified starch and polyamine anti sloughing inhibitor, the viscosity and shear force of drilling fluid increase, and the water loss reduction effect is good. The application of the high temperature modified starch polyamine drilling fluid technology to the second spud of a deep shale gas horizontal well in the Yongchuan area of China has effectively reduced the rate of complex incidents such as sticking, mud bagging, and reaming when encountering obstacles in the second spud section. The time taken to trip an obstacle in the second spud of a single well has been reduced from 2–3 d in the previous period to 3–10 h, and the reuse rate of the aged slurry in the second spud is 100%.

## 5. Conclusions

In this paper, we discuss the mechanisms that inhibit hydration expansion in clay minerals, evaluate the adsorption properties of anti-collapse inhibitors on the surface of clay grains and their effect on the morphology of bentonite, and evaluate the rheology, high temperature filtration and stability of optimized water-based drilling fluids. The optimization idea of controlling the rheological property and water loss of drilling fluid with high temperature resistant modified starch and strengthening the inhibition performance of drilling fluid with “KCl + polyamine” was proposed, forming a high temperature-resistant modified starch polyamine anti-sloughing drilling fluid system with stable performance and strong plugging and strong inhibition.

The mechanism by which polyamine inhibitors and anti-collapse agents inhibit the hydration, expansion, and dispersion of clay-forming minerals is that polyamines enter the sodium bentonite layer and squeeze some water molecules out of the layer, reducing the interlayer water content and changing the surface morphology of the bentonite. After the monolayer adsorption of the polyamine inhibitor antiskid agent on the surface of the clay particle, the hydrophobic groups on the polyamine inhibitor antiskid agent molecular chain partially cover the clay surface. Experimental measurements have shown that the adsorption at the clay surface is between 66 and 74, which makes the clay surface structure longer, reduces the hydrophilicity of the clay, enhances the hydrophobicity, prevents the entry of water molecules, and further inhibits the hydration expansion of the clay.

The modified starch material was obtained by grafting starch onto a different type of high-performance polymer monomer. The performance of the water-based drilling fluid system was evaluated experimentally with the addition of modified starch. The experimental results showed that its temperature resistance reached 180 °C, and it had excellent salt resistance, calcium resistance and inhibition properties.

A water-based drilling fluid system with high temperature-resistant modified starch as the primary fluid loss reducer and KCl polyamine as the primary inhibitor was developed, which is characterized by high temperature resistance, environmental protection, strong sticking properties, strong inhibition, and stability properties. The results of the mini-scale field tests show that the optimized drilling fluid has increased viscosity and shear force, and the water loss reduction effect is significant. The application of high temperature-resistant modified starch polyamine drilling fluid technology in the second spudding of deep shale gas horizontal wells in Yongchuan area in China effectively reduces the occurrence rate of drill pipe sticking, mud bag, reaming, and other complex accidents when encountering obstacles in the second spudding section, saves drilling costs, and considerably supports the excellent and rapid drilling and completion technology of horizontal wells.

## Figures and Tables

**Figure 1 molecules-27-08936-f001:**
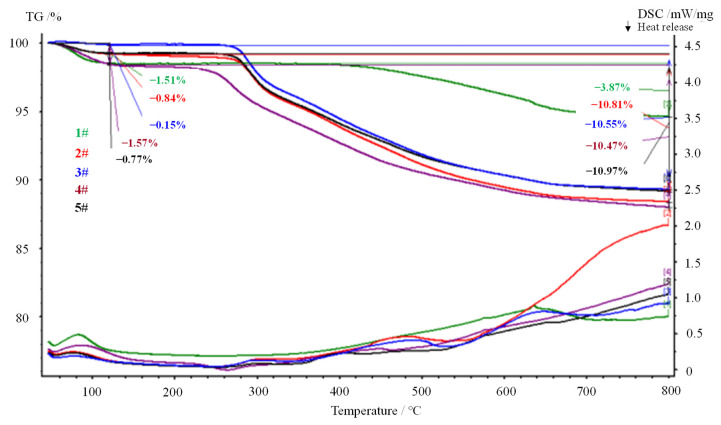
Thermogravimetric curves of polyamine inhibitors.

**Figure 2 molecules-27-08936-f002:**
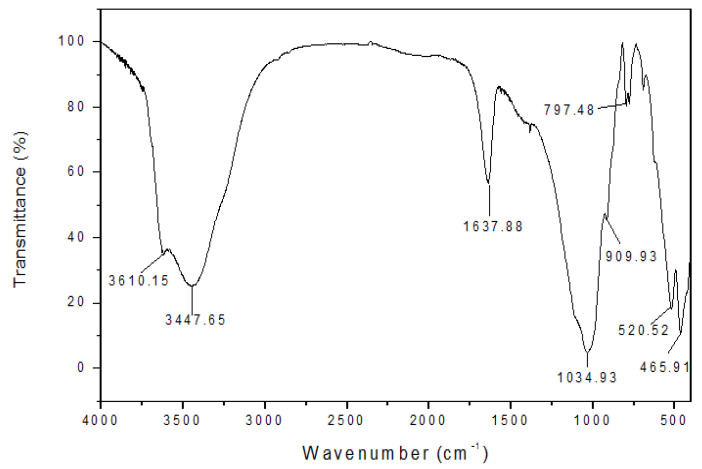
Infrared spectrogram of sodium bentonite for laboratory evaluation.

**Figure 3 molecules-27-08936-f003:**
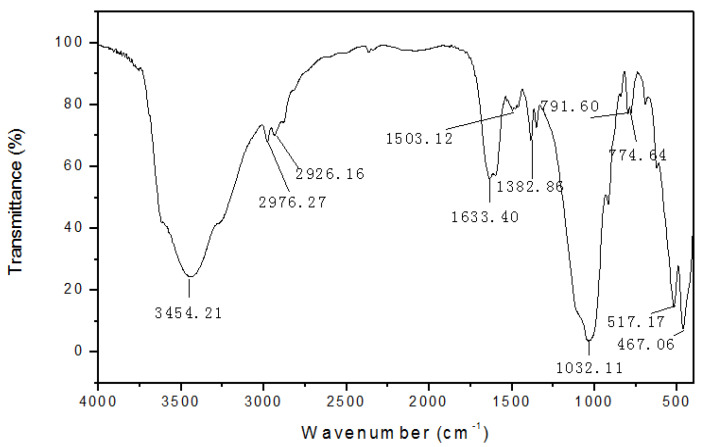
Infrared spectra of polyamine modified soil.

**Figure 4 molecules-27-08936-f004:**
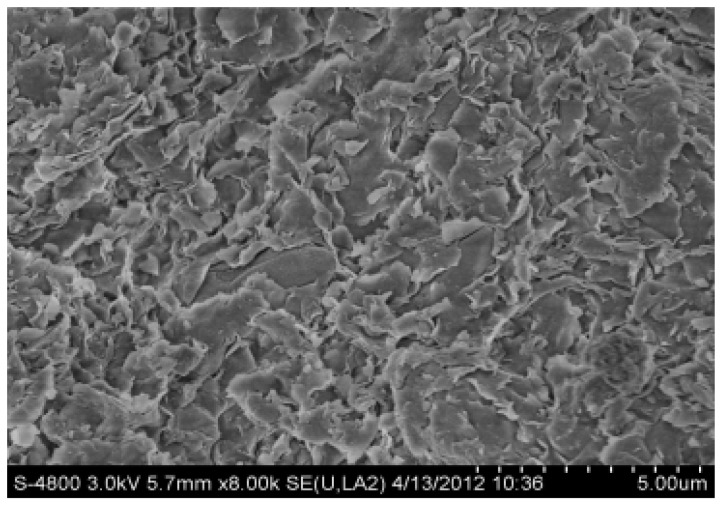
SEM of bentonite soaked in water.

**Figure 5 molecules-27-08936-f005:**
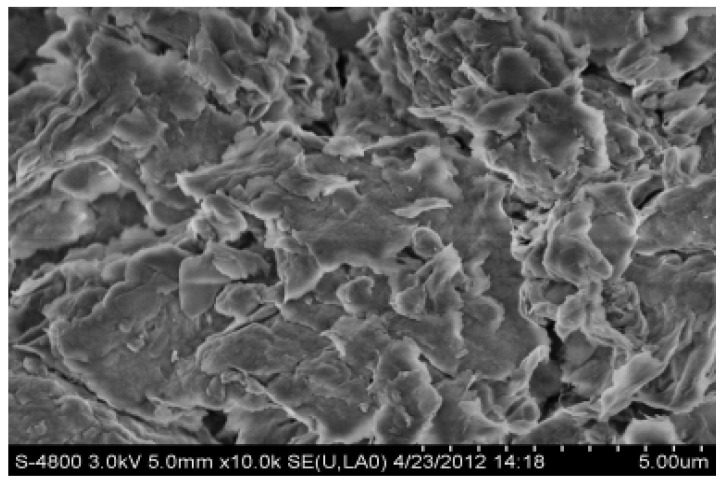
SEM of bentonite soaked in polyamine solution.

**Figure 6 molecules-27-08936-f006:**
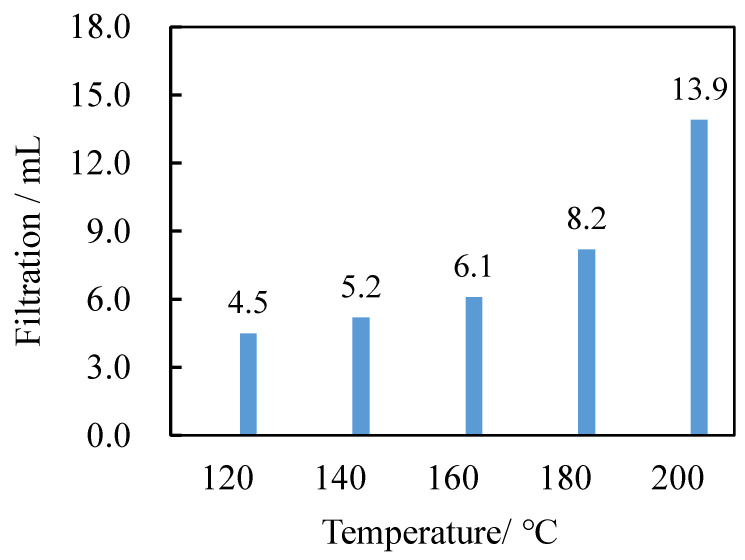
Experimental results of temperature resistance evaluation of modified starch.

**Figure 7 molecules-27-08936-f007:**
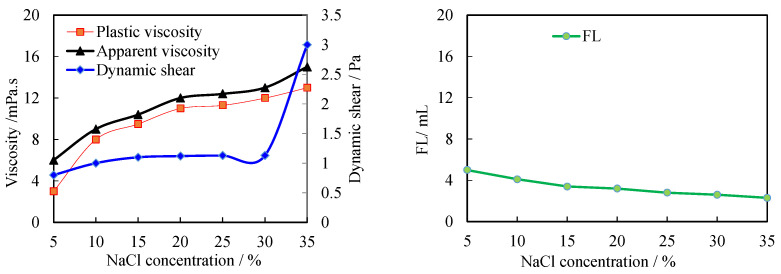
Salt resistance test results of modified starch.

**Figure 8 molecules-27-08936-f008:**
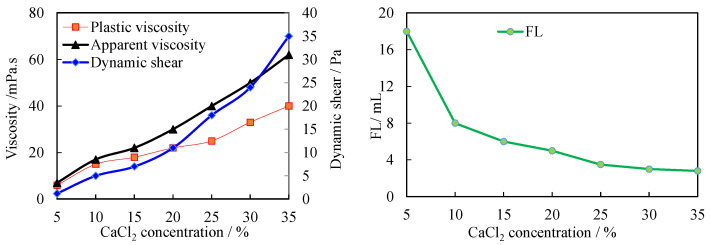
Evaluation of salt resistance of modified starch.

**Table 1 molecules-27-08936-t001:** Adsorption capacity of polyamine inhibitor on clay.

No.	Sample	Weight Loss Rate after 100 °C/%	Inhibitor Weight Loss Rate/%	Adsorption Capacity/mg/g
1	Soil for experimental evaluation	3.91	-	-
2	Polyamine modified soil	10.83	6.97	73.94

**Table 2 molecules-27-08936-t002:** Fundamental frequency vibration and main absorption peaks of infrared spectral activity of sodium bentonite.

Attribution of Absorption Peak	Absorption Peak Position/cm^−1^
Structural water OH bond expansion vibration absorption	3618
Water molecule stretching vibration absorption band	3416
Flexural vibration absorption band of water molecules	1627
Si-O bond telescopic vibration absorption ban	1040
OH bond bending vibration absorption band	915 and 887
Si-O-R^3+^ vibration absorption band	520 and 477

**Table 3 molecules-27-08936-t003:** Experimental evaluation of rheology, water loss and aging test.

Temperature /°C	Time /h	FL_API_ /mL	FL_HTHP_ /mL	pH	G10″/G10′	Φ_600_/Φ_300_	PV/mPa.s	YP/Pa	Rolling Recovery Rate /%	Linear Expansion Rate/%	
										8 h	16 h
80	16 h	3	11	10	2/12	95/59	36	11.5	98.0	11.44	14.71
	32 h	2.6	10.5	10	7/12	92/64	28	18	97.5	10.9	13.77
	64 h	3.6	11.8	10	2/8	78/52	26	13	93.5	12.66	14.67
120	16 h	2	8.4	10	4/15.5	93/59	34	12.5	97.0	10.12	13.07
	32 h	3.2	8	10	2/5	97/58	39	9.5	99.0	11.64	15.09
	64 h	3.0	18	10	1.5/3	94/57	37	10	93.5	11.81	15.33
140	16 h	2.6	11.4	10	2/2.5	92/55	37	9	98.5	13.24	13.72
	32 h	2.8	10.5	10	1.5/3	109/66	43	11.5	99.0	12.27	15.68
	64 h	3.6	11	10	0/0.5	51/29	21	4.5	99.5	12.37	14.9
160	16 h	2.6	11	10	0.5/0.5	55/30	25	2.5	99.5	10.73	13.78
	32 h	3.4	12.5	10	0/0.5	55/32	23	4.5	99.5	13.44	18.09
	64 h	4.2	15	10	0/0.5	31/18	13	2.5	97.0	8.11	11.36

**Table 4 molecules-27-08936-t004:** Evaluation of salt resistance of polyamine potassium chloride system.

NaCl Dosage /%	FL /mL	Cl^−^ /ppm	G10″/G10′	Rheological Parameters			FL_HTHP_ /mL	Shale Inhibition /%	
				Φ_600_/Φ_300_	PV /mPa.s	YP /Pa		8 h	16 h
0%	2.6	34,000	2/2.5	92/55	37	9	10.4	13.24	13.72
5%	1.8	65,000	2/2.5	93/56	37	8.5	12.0	10.37	13.75
10%	1.8	100,000	1/5.5	102/62	40	11	11.6	8.6	11.97
15%	5.4	135,000	1/3	112/69	43	13	14.8	14.89	17.03

**Table 5 molecules-27-08936-t005:** Evaluation of calcium resistance.

CaCl_2_ Dosage /%	Ca^2+^ /ppm	FL /mL	G10/G10	Rheological Parameters			FL_HTHP_ /mL	Shale Inhibition /%	
				Φ_600_/Φ_300_	PV/mPa.s	YP /Pa		8 h	16 h
0%	0	2.6	2/2.5	92/55	37	9	10.4	13.24	13.72
0.4%	1500	2.4	4/7	104/65	39	13	10.2	12.22	14.83
0.8%	3000	2.4	7/10	114/73	41	16	11.4	11.54	15.63
1.2%	4200	3.2	9/12	123/76	46	15	11.8	19.42	26.34
1.6%	5500	4.0	12/14	144/93	51	21	12.4	22.67	28.46

**Table 6 molecules-27-08936-t006:** Evaluation of anti-poor soil pollution ability.

Inferior Soil Dosage /%	FL /mL	G_10_″/G_10_′	Rheological Parameters					FL_HTHP_ /mL
			Φ_600_/Φ_300_	Φ_200_/Φ_100_	Φ_6_/Φ_3_	PV /mPa.s	YP /Pa	
0	2.6	2/2.5	92/55	40/24	5/4	37	9	11.4
5	2.4	4/6	102/62	51/29	7/4	40	11	11.0
10	2.2	5/9	122/76	56/33	10/7	46	15	10.6
15	1.8	5/10	126/79	61/37	10/8	47	16	9.2
20	1.8	6/11	142/91	69/46	12/9	51	20	8.6

**Table 7 molecules-27-08936-t007:** Optimized performance parameters of the modified starch polyamine anti-collapse water-based drilling fluid tested on wellsite.

Type	General Performance						HTHP		Rheological Properties		Solid Content			K_f_ /45′	K^+^ /mg/L
Parameter	FV/s	FL /mL	K /mm	pH	G10″ /Pa	G10′ /Pa	FL /mL	K /mm	PV /mPa.s	YP /Pa	Vs /%	Cb /g/L	Cs /%		
Test result	50~70	≤4	≤0.5	9~10	5~10	12~16	≤10 (140 °C)	≤2	22~36	10~16	≤30	20~35	≤0.2	≤0.15	15,000

## Data Availability

Not applicable.

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
