# Peer review of "Optimization of High Temperature-Resistant Modified Starch Polyamine Anti-Collapse Water-Based Drilling Fluid System for Deep Shale Reservoir"

_molecules, 2022, doi:10.3390/molecules27248936_

Round 1

Reviewer 1 Report

The article has a probability of the acceptance to the journal while it must be revised. It has some poor parts for accepting to the journal as follows.

1) The explanation about Figure 6 is poor because it is not clearly agreed with the conclusion: temperature resistance of 180 0C.  

2) The drawing and presentation of Figures 7 and 8 are poor because comparisons between NaCl and CaCl2 or among two kinds of viscosity, dynamic shear and FL are indicated by different scales between and among them. The unit scale of X-axis should be the same between the left-side figure and right-side figure of Figures 7 and 8.

4) The evaluation results based on the data shown in Table 3-7data are too complicated to be understood by the journal readers. The revisions (for example, making one summarized table from Table 3-7 which are moved to the supplemental data) are required to make their understanding improve. 

Author Response

Comment 1: The explanation about Figure 6 is poor because it is not clearly agreed with the conclusion: temperature resistance of 180 ℃.

Response 1: Thanks for your valuable advice. This experimental evaluation method is based on the national standard of China: GB/T 16783.1-2014, Petroleum and natural gas industries - Field testing of drilling fluids - Part 1: Water based drilling fluids, which conforms to the API international standard.

If the filtration loss of drilling fluid is kept below 12mL, it indicates that this filtration reducing additive has good filtration reducing performance and meets the national industry standard. It can be seen from Figure 6 that when the experimental temperature is 120-180 ℃, the filter loss does not exceed 12mL, and the increase of the filter loss is relatively slow with the increase of the temperature. However, when the temperature reaches 200 ℃, the filter loss exceeds 12mL, and the increase trend of the filter loss is stronger than that of the previous filter loss. Therefore, the analysis shows that the filter loss at this temperature is not up to the standard, indicating that the performance of the additive is poor when the temperature is 200 ℃, so the temperature resistance of this material can reach 180 ℃.

Comment 2: The drawing and presentation of Figures 7 and 8 are poor because comparisons between NaCl and CaCl2 or among two kinds of viscosity, dynamic shear and FL are indicated by different scales between and among them. The unit scale of X-axis should be the same between the left-side figure and right-side figure of Figures 7 and 8.

Response 2: Thanks for your valuable advice. Figure 7 and Figure 8 show the salt resistance test results of the modified starch added to the water-based drilling fluid base fluid. The performance parameter values of the modified starch at different NaCl concentrations to reduce the filtration of drilling fluid were tested, including AV, PV, YP and static filtration.

We have redrawn Figure 7 and Figure 8, unifying the abscissa values (the addition of NaCl and CaCl2) and ordinate values of the two figures to facilitate numerical comparison.

Comment 3: The evaluation results based on the data shown in Table 3-7data are too complicated to be understood by the journal readers. The revisions (for example, making one summarized table from Table 3-7 which are moved to the supplemental data) are required to make their understanding improve.

Response 3: Thanks for your valuable advice. The experimental conditions in Table 3 and Table 4 of the original text are the same, so we will combine Table 3 and Table 4 into one table so that readers can easily understand them. Table 5, 6 and 7 respectively show the salt resistance, calcium resistance and poor soil pollution resistance of drilling fluid under different experimental conditions. With different experimental conditions, the dosage of single agent is different, and the fluid parameters tested are also different. If combined and unified, I think the experimental results will be confused, and it is difficult to make a correct evaluation of the three properties. Therefore, it is recommended to retain the independence of the three tables.

Reviewer 2 Report

1. Is it possible to propose an optimization approach? This is because this article is more like a report.

2. Sec.4.5 Application is very simple. It is suggested that more details should be introduced. There should be some comparision results in the application phase.

3. The conclusions are more like the results. 

4. Some figures are not clear, such as Fig.1.

Author Response

Comment 1: Is it possible to propose an optimization approach? This is because this article is more like a report.

Response 1: Thanks for your valuable advice. We reorganized and revised the content of the manuscript, and added Section 3.1 Action mechanism of modified starch, and the mechanism of modified starch to reduce filtration was introduced.

Comment 2: Sec.4.5 Application is very simple. It is suggested that more details should be introduced. There should be some comparision results in the application phase.

Response 2: Thanks for your valuable advice. We have revised the content of Sec. 4.5, added a small field test of drilling fluid performance parameters, analyzed the test parameters, and described the main parameters for performance improvement to prove the improvement of the optimized water-based drilling fluid.

Comment 3: The conclusions are more like the results.

Response 3: Thanks for your valuable advice. We rewrote the conclusion of the manuscript, strengthening the inhibition performance of drilling fluid with "KCl+ polyamine" was proposed, forming a high temperature resistant modified starch polyamine an-ti-sloughing drilling fluid system with stable performance and strong plugging and strong inhibition.

Comment 4: Some figures are not clear, such as Fig.1.

Response 4: Thanks for your valuable advice. We revised Fig.1 in the manuscript.

Round 2

Reviewer 2 Report

I have no further comments. It can be accepted.